# Anticoagulation Control and Major Adverse Clinical Events in Patients with Operated Valvular Heart Disease with and without Atrial Fibrillation Receiving Vitamin K Antagonists

**DOI:** 10.3390/jcm12031141

**Published:** 2023-02-01

**Authors:** Hanis H. Zulkifly, Daniele Pastori, Deirdre A. Lane, Gregory Y. H. Lip

**Affiliations:** 1Fakulti Farmasi, Puncak Alam Campus, Universiti Teknologi MARA, Puncak Alam 42300, Malaysia; 2Cardiology Therapeutics Research Group, Universiti Teknologi MARA, Puncak Alam 42300, Malaysia; 3Emergency Medicine Unit, Department of Clinical, Internal, Anesthesiologic and Cardiovascular Sciences, Sapienza University of Rome, 00185 Rome, Italy; 4Liverpool Centre for Cardiovascular Science at University of Liverpool, Liverpool John Moores University, Liverpool Heart & Chest Hospital, Liverpool L14 3PE, UK; 5Department of Clinical Medicine, Aalborg University, DK-9100 Aalborg, Denmark

**Keywords:** anticoagulation control, vitamin K antagonist, atrial fibrillation, operated valvular heart disease, major adverse clinical events, all-cause death

## Abstract

Background: Good quality anticoagulation among patients with operated valvular heart disease is needed to reduce ischaemic and thromboembolic complications. There is limited evidence regarding factors that affect anticoagulation control in patients implanted with mechanical or tissue prosthetic valve(s). Aim: To examine the quality of and factors that affect anticoagulation control, major adverse clinical events and all-cause death in operated valvular heart disease patients with and without atrial fibrillation who are receiving a vitamin K antagonist. Methods: Quality of anticoagulation were retrospectively assessed among 456 operated valvular heart disease patients [164 (36%) with AF and 290 (64%) without AF] via time in therapeutic range (TTR) (Rosendaal method) and percentage of INRs in range (PINRR) over a median of 6.2 (3.3–8.5) years. VHD was defined by the presence of a mechanical or tissue prosthetic valve at the mitral, aortic, or both sites. Results: Mean age 51 (14.7), 64.5% men. Most (96.1%) had a mechanical prosthesis and 64% had aortic valve replacement. Overall, mean TTR was 58.5% (14.6) and PINRR was 50.1% (13.8). Operated valvular heart disease patients with AF had significantly lower mean TTR and PINRR (TTR: 55.7% (14.2) vs. 60.1% (14.6); *p* = 0.002, respectively, PINRR: 47.4% (13.5) vs. 51.6% (13.7); *p* = 0.002, respectively), and a lower proportion of TTR ≥ 70%, despite a similar number of INR tests compared to those without AF. Predictors of TTR < 70% were female sex, AF and anaemia/bleeding history. Significantly higher proportions of operated valvular heart disease patients with AF died (20.7% vs. 5.8%; *p* < 0.001), but ≥1 MACE rates were similar between the two groups. Conclusions: Operated valvular heart disease patients with AF at baseline have poorer anticoagulation control compared to those without AF. The presence of concomitant AF, anaemia/bleeding history and female sex independently predicted poor TTR. Stringent INR monitoring is needed to improve anticoagulation control and prevent major adverse clinical events in patients with operated valvular heart disease.

## 1. Introduction

Vitamin K antagonists (VKAs) are the only licensed anticoagulant drugs for thromboprophylaxis of patients undergoing mechanical heart valve replacement [1]. The target international normalised ratio (INR) for atrial fibrillation (AF) patients is 2.0–3.0 [1,2,3] whereas the INR targets for patients with valvular heart disease (VHD) post-surgery varies depending on factors such as patient risk factors, (example: valve site, previous thromboembolism, AF, mitral stenosis and left ventricular ejection fraction (LVEF) < 35%) type of valve, and the thrombogenicity of the prosthesis [1,2,3]. The 2021 European Society of Cardiology (ESC) guidelines on the management of valvular heart disease [4] recommend maintaining a median INR value (e.g., median INR of 3.0 instead of a range 2.5–3.5 in those receiving medium thrombogenicity valve (e.g., bileaflet valves) who do not have patient-related risk factor (e.g., previous thromboembolism, AF)) rather than a target INR range to prevent considering extreme values within the target range. They also recommend a higher median INR value for patients with ≥1 risk factor than those without any of these risk factors [1]. The newer and more common types of valve have low valve thrombogenicity; however, the data on the rate of valves thrombogenicity is limited as they are rates are influenced by patient related risk factors and study design [5]. Patients with risk factors receiving newer types of valves are recommended to achieve a median target INR of 3.0 compared to those without risk factors (including AF), where the target INR is lower (2.5) [1,2,3].

Various studies [6,7,8,9,10,11,12] conducted between 2002 and 2021 have investigated anticoagulation control after valve replacement; two studies [8,10] used anticoagulation variability while the others [6,7,9,10] used time in therapeutic range (TTR). Therefore, the objectives of this study were to investigate anticoagulation control measured using TTR (Rosendaal method) and the PINRR method among operated patients with VHD both with and without AF and to investigate the predictors of poor anticoagulation control (TTR < 70%) and the prevalence of major adverse clinical events (MACE).

## 2. Materials and Methods

This study conducted a retrospective analysis of patients from a single centre with operated VHD receiving VKA therapy after valve replacement at one acute NHS Trust in the UK. Data were collected from 1 November 2017 to 31 March 2018. Anticoagulation management software was used to identify VHD patients receiving VKA therapy. This study was considered a service evaluation by the Research and Development department and therefore did not require Research Ethical Committee (REC) approval, however local Research and Development (R&D) approval was obtained.

### 2.1. Patient Selection

A list of patients with VHD receiving VKA therapy (*N =* 604) was generated from the anticoagulation management software. A total of 148 (24.5% 148/604) patients were excluded because of: (i) VHD but without surgical intervention (*n* = 38) [mitral stenosis (*n* = 22), aortic stenosis (*n* = 2), mitral regurgitation (*n* = 2), mitral valve repair (*n* = 4), valvuloplasty (*n* = 6), and valvulotomy (*n* = 2)]; (ii) incomplete INR results (*n* = 3) and (iii) incomplete medical information (*n* = 107). The final cohort consisted of 456 VHD patients who had surgical intervention of the affected valve(s) and were prescribed VKA therapy post-surgery. They were stratified into those with and without AF (Figure 1).

### 2.2. Procedure

All baseline characteristics and clinical information, including medical history, medication and laboratory tests, were collected from the point when VKA was initiated after surgical replacement of the valves (i.e., mechanical and tissue valve replacement). Information on outcomes i.e., INR results and MACE were collected from 1 February 2009 until 31 January 2018, for a median of 6.2 (3.3–8.5) years.

### 2.3. Time in Therapeutic Range (TTR)

INR values for VKA therapy were collected from the hospital anticoagulation management software for patients with at least three INR values in a year starting from 1 February 2009 to 31 January 2018. The year 2009 was chosen as this is when INR readings became consistently available in the hospital databases. The quality of anticoagulation control was calculated using the Rosendaal and (using linear interpolation method between two consecutive INR values) the PINRR methods [13]. TTR and PINRR were calculated based on each patient’s individual target INR range as determined by the surgeon at the time of valve implantation; thus, INR ranges differed between patients. TTR and PINRR were further dichotomised into TTR ≥ 70% and <70% and PINRR ≥ 70% and <70%, with TTR and PINRR ≥ 70% reflecting optimal anticoagulation control based on the ESC guideline [14,15]. The proportions of sub-therapeutic INRs (INRs below the target range), supra-therapeutic (INRs above target range) and patients with at least one INR > 5.0 or >8.0 were also collected, as these are also indicators of poor anticoagulation control [16]. Furthermore, the risk of bleeding also increases when the INR exceeds 4, and the risk rises sharply with values > 5 and >8 [17]. The follow-up period was defined as the duration of VKA monitoring i.e., from the start date of INR collection (1 February 2009) until 31 January 2018.

### 2.4. Patient Demographics and Clinical Factors

Patient’s age was calculated based on the date of their first valve surgery. Other demographic information was collected from the clinical data archive (CDA), such as gender, ethnicity, information regarding smoking status and alcohol intake, and other comorbidities, medication history and laboratory parameters were obtained as close as possible to the date (within one month) of VKA initiation after the first valve surgery. Diagnosis of AF at baseline was defined as the presence of known AF at the time of surgery, or a post-operative diagnosis of AF. Calculation of the CHA_2_DS_2_-VASc, HAS-BLED and SAMe-TT_2_R_2_ scores were made based on baseline information.

### 2.5. Major Adverse Clinical Events (MACE)

Information on adverse clinical outcomes were collected from the EHR covering the same timeframe as the INR collection, i.e., from the point/date where INR was consistently available in the system until 31 January 2018. Major adverse clinical events (MACE) of interest were stroke/transient ischemic attack (TIA)/systemic embolism, bleeding (combination of major bleed and clinically relevant non-major bleed), cardiovascular (CV) hospitalisation, death and a composite (≥1) of any MACE. Stroke and systemic embolism were combined as thromboembolic events (TE). Major bleeding and clinically relevant non-major bleeding (CRNMB) was classified according to the ISTH criteria [18]. Major bleeding and CRNMB were combined as bleeding events. In this study, the cause of death was specified as CV death when specific information was available. Where cause of death was unavailable, death was classified as all-cause death.

### 2.6. Statistical Analysis

After performing normality tests, all normally distributed data were expressed as mean, and non-normally distributed data as median. Demographic and clinical characteristics of patients with categorical data were compared with chi-squared or Fisher’s exact test where appropriate and are reported as counts and percentage. Independent *t*-tests were used to compare the means of continuous data for normally distributed data; the Mann–Whitney tests were used for data that was not normally distributed. Univariate and multivariate logistic regression analyses were performed to investigate the predictors of poor TTR (TTR < 70%). Six models incorporating the predictors of poor TTR were developed via multivariate logistic regression analysis in the overall cohort of VHD patients. The relationship between the presence of AF and adverse clinical outcomes among VHD patients were investigated in an exploratory analysis using the chi-squared test and are reported as counts and percentage. A log-rank test was performed for AF categories and Kaplan–Meier curves were used to report the differences in survival and ≥1 MACE between the sub-groups. All analyses were conducted using SPSS version 23.0 [19], with *p*-values < 0.05 considered statistically significant.

## 3. Results

Among the 456 patients with operated VHD, 164 (36.0%) had AF at baseline. Of these, 23.2%, 19.5% and 46.3% had paroxysmal, persistent and permanent AF, respectively. The overall mean age of VHD patients at the time of their first valve surgery was 51 years (14.7), most were male (64.5%), of white ethnicity (65.2%), with a mechanical prosthesis (96.1%), and the most common operation was aortic valve replacement (64%) (Table 1). Patients with operated VHD with AF were significantly older [mean age 56.6 (13.3) vs. 48.0 (15.0); *p* < 0.001], more likely to be female (48.2% vs. 28.4%; *p* < 0.001), to receive a tissue prosthesis (8.5% vs. 1.4%; *p* < 0.001), and to have had the mitral valve (41.5% vs. 14.4%; *p* < 0.001) or both mitral and aortic (20.7% vs. 6.8%; *p* < 0.001) valves replaced. Patients with operated VHD and AF were also more likely to have concomitant heart failure, hypertension, and pulmonary disease and were likely to be prescribed diuretics, amiodarone and digoxin; they also had higher mean CHA_2_DS_2_-VASc [2.6 (1.5) vs. 1.7 (1.3); *p* < 0.001] and HAS-BLED scores [1.8 (1.1) vs. 1.5 (1.2); *p* = 0.014] compared to patients with operated VHD without AF (Table 1). No significant differences were evident in the SAMe-TT_2_R_2_ score among operated VHD patients with and without AF.

### 3.1. Quality of Anticoagulation Control

As shown in Table 2, higher INR target ranges [INR 3.0–4.0; 41.4%] were used more often to maintain effectiveness and safety of VKA therapy in the overall population of patients with operated VHD. The overall mean TTR and PINRR for the cohort was 58.5 (14.6) and 50.1 (13.8) respectively; only 98 patients (21.5%) achieved the optimal TTR target (TTR ≥ 70%) during a median of 6.2 (3.3–8.5) years of follow up.

Operated VHD patients with AF had a significantly higher INR target range and lower mean TTR and PINRR [mean TTR 55.7 (14.2) vs. 60.1 (14.6); *p* = 0.002 respectively; mean PINRR 47.4 (13.5) vs. 51.6 (13.7); *p* = 0.002 respectively] (Table 2 and Figure 2a), lower proportions with optimal anticoagulation control (TTR ≥ 70%) (14.0% vs. 25.7%; *p* = 0.004) (Figure 2a) and higher proportions with sub-therapeutic INRs (28.4% vs. 23.4%; *p* < 0.001) (Figure 2b) despite a similar number of INR tests compared to operated VHD patients without AF. There was no significant difference in INRs above the therapeutic range or the proportions of patients with one or more incident of INR > 5.0 or >8.0 between those with and without AF (Table 2).

### 3.2. Predictors of Poor Anticoagulation Control

Models 1–6 in Table 3 present the independent factors that predict poor TTR after adjustment for demographic and clinical variables (Appendix A unadjusted analysis). Being female, the presence of AF at baseline, and anaemia/bleeding history, were consistently present in 4 of the 6 models predicting poor TTR. The HAS-BLED score, which also contains anaemia/bleeding history, also predicted poor TTR in 2 of the 6 models (models 4 and 6).

### 3.3. Major Adverse Clinical Events (MACE)

Overall, there were 31 TE events, 113 bleeding events, 123 CV hospitalisations, 51 deaths and 316 patients experienced ≥1 MACE. There were no significant differences in TE, bleeding, CV hospitalisation and ≥1 MACE between those with and without AF. However, significantly higher proportions of patients with operated VHD and AF died (all-cause death (20.7% vs. 5.8%; *p* < 0.001); CV death (7.3% vs. none; *p* < 0.001) and non-CV death (13.4% vs. 5.8%; *p* = 0.009)) compared to those with operated VHD without AF (Table 4). In survival analyses, operated VHD patients with AF had a significantly higher risk of all-cause death compared to those without AF (log-rank: 21.570; *p* < 0.001; Figure 3). Higher proportions of patients died (13.1% vs. 4.1%; *p* = 0.011) and experienced ≥1 MACEs (42.7% vs. 27.6%; *p* = 0.006) when their TTR was <70% compared to those with TTR ≥ 70% (Appendix A). In survival analyses, patients with TTR < 70% had a significantly higher risk of all-cause death (log-Rank: 5.845, *p* = 0.016; Figure 4a) and ≥1 MACEs. (log-rank: 7.541, *p* = 0.006; Figure 4b).

## 4. Discussion

This study has three main findings. First, the quality of anticoagulation control was significantly lower in operated VHD patients with AF at baseline compared to those without AF at baseline, using both the Rosendaal and PINRR methods. Second, females, the presence of AF and anaemia/bleeding history significantly predicted poorer anticoagulation control in the overall cohort. Third, the rate of death was significantly higher in those with operated VHD with AF compared to operated VHD patients without AF. As far as we are aware, this is the first study assessing the quality of anticoagulation control among operated VHD patients stratified by the presence of AF at baseline.

In the present study, the mean TTR was 58.5% (14.6%) and less than a quarter of the cohort achieved optimal TTR (TTR ≥ 70%), reflecting poor anticoagulation control among operated VHD patients. There is a paucity of literature on the quality of anticoagulation control among operated VHD patients, especially those with AF. Various studies are available assessing TTR among VHD patients [6,7,9,10,11,12]. The Swedish groups [9] examined TTR among 534 patients [9], and 4687 patients [10] with mechanical heart valves and reported a mean TTR of 71.3% [9] and 72.5% [10] respectively, which is higher than the mean TTR in the present study. In contrast, four recent studies [6,7,11,12], conducted in Italy (*N =* 2111–2357) [6,11,12] and Denmark (*N =* 659) [7], reported a median (IQR) TTR of 60% (47–74%) and 54.9% (39.0–72.9%), respectively; this is comparable to the TTR in the current study. The findings of the Italian and Danish studies [6,7,11,12] and the current study show sub-optimal quality of anticoagulation control among operated VHD patients. In contrast, the two Swedish studies [9,10] showed optimal anticoagulation control among operated VHD patients, although this could be explained by the fact that generally, Sweden [9] is known to have excellent anticoagulation management, resulting in better TTR compared to other countries [20,21]. This again reinforces one important message: the difficulties in maintaining INR levels at the therapeutic range among anticoagulated operated VHD patients and it is worst in those with underlying AF compared to those without AF at baseline. Furthermore, in this cohort of operated VHD patients with AF were more likely to have VHD at mitral site, which requires a higher INR target than either NVAF patients or patients with VHD at aortic site; this target may be more difficult to achieve and maintain. This is more worrying in patients with concomitant AF, as AF patients with VHD carry an even higher risk of TE complications (5–62%) [22] than patients with NVAF (0–18%) [22].

In logistic regression analyses, after adjusting for demographics and clinical variables, being female, the presence of AF and anaemia/bleeding history consistently predicted poor TTR in four of the six models. In addition, the HAS-BLED (which also includes anaemia/bleeding history) score also predicted poor TTR (<70%) in two of six models. The finding that being female predicts poor TTR is consistent with other non-valvular AF studies [23,24,25,26,27,28,29]. One large American study [30] evaluating medication use and adherence among 16.0 million women and 13.5 million men showed that women were more likely to be non-adherent to their diabetic (35.4% vs. 32.5%; *p* < 0.0001) and antihypertensive (25.8% vs. 24.8; *p* < 0.0001) medications compared to men, and also speculated that their higher self-neglect, compared to men, resulted from having more complex medications regimes, more side effects and more responsibilities [30]. Furthermore, in this study, the majority of the operated VHD females also had AF at baseline, which is also a predictor of poor TTR. Operated VHD patients with AF are older and have multiple comorbidities with complex disease management, which might contribute to the lower quality of anticoagulation control [11,12,23,25,27,31]. Lastly, history of anaemia/bleeding among operated VHD patients was also an independent predictor of poor TTR consistent in another study among non-valvular AF patients [32]. It may be that these patients were managed more cautiously in terms of VKA dosing. Although information on the dosage of VKAs used was not available, a lower dosage may have been used in this group of patients because of the fear of bleeding complications leading to the risk of suboptimal anticoagulation control in this population. No other studies have investigated the predictors of TTR specifically among operated VHD patients, so comparison with other studies regarding the predictors in this population could be made. However, Poli et al. [6] investigated the predictors of TE among mechanical heart valves patients and showed that AF, history of TE and prosthesis at mitral position were associated with TE complications [6].

During a median follow up of 6.2 years, 11% of the patients died, which was also higher than that reported by Poli et al. [6] (7.4% deaths); this might be explained by differences in the demographic and clinical characteristics of the cohorts. There were more males, patients from ethnic minority groups, smokers/ex-smokers and a higher disease burden (stroke/TIA, diabetes, vascular disease and anaemia) in the present study, which could potentially contribute to the differences in the mortality rate. The proportions of operated VHD patients who had a TE, bleeding event, CV hospitalisation and ≥1 MACEs were similar among those with and without AF at baseline. Nevertheless, all-cause death (including CV and non-CV related death) was significantly higher among those with AF compared to those without AF, indicating that in this cohort, patients with operated VHD and AF have a worse prognosis than those without AF.

### Strengths and Limitations

This is the first study investigating anticoagulation control in the UK among operated VHD patients stratified by the presence of AF at baseline (obtained from the post-operative notes). Although it is limited by the relatively small sample size (with approximately one-sixth of the eligible cohort excluded because of missing medical information), it provides some insights on anticoagulation control among operated VHD patients both with and without AF. Studies investigating anticoagulation control among VHD patients are lacking, thus the information gained from this study adds to the limited current literature. In addition, anticoagulation control was assessed for 6.2 years, reflecting long term anticoagulation control among VHD patients.

This study is limited by its retrospective, single centre design and the small number of operated VHD patients included, so caution must be applied as the findings might not be transferable to other settings. There is no information on the proportion of pregnancies, the doses of VKAs and type of valve inserted in the patients, whether patients were offered patient self-monitoring (PSM) or home monitoring service, distance to anticoagulation clinic, level of education, drugs and food interaction or genetic information, all of which could impact the quality of anticoagulation control. Additionally, analyses pertaining to major adverse clinical outcomes were exploratory in nature.

## 5. Conclusions

Operated VHD patients with AF at baseline have poorer anticoagulation control compared to those without AF at baseline. The presence of concomitant AF, anaemia/bleeding history and female gender independently predicted poor TTR. Stringent INR monitoring is needed to improve anticoagulation control and prevent major adverse clinical events in patients with operated VHD.

## Figures and Tables

**Figure 1 jcm-12-01141-f001:**
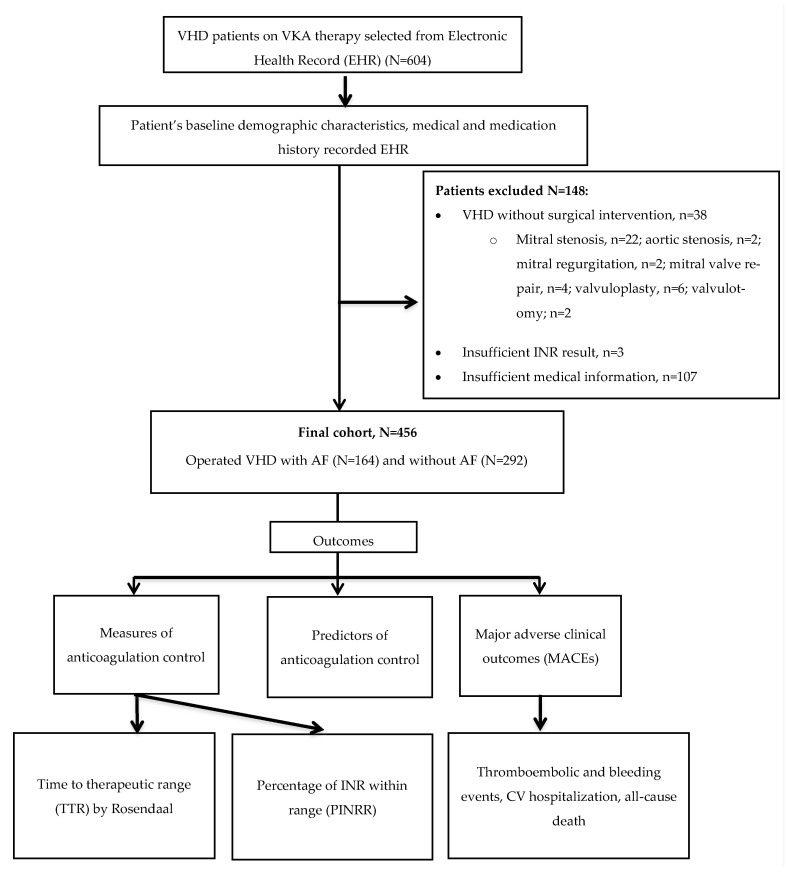
Study design and flow chart.

**Figure 2 jcm-12-01141-f002:**
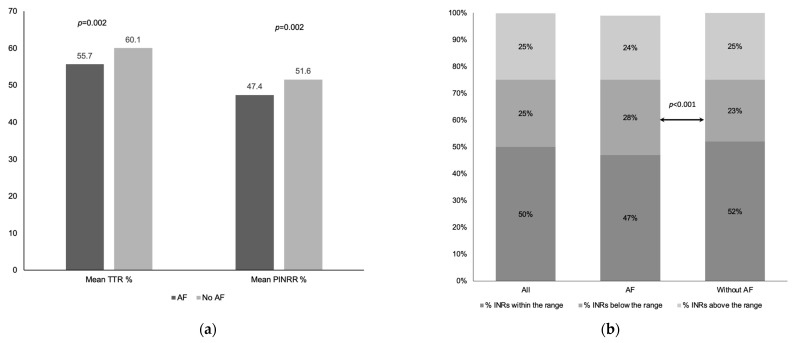
(**a**) Mean TTR and PINRR among operated valvular heart disease, with and without AF; (**b**) Percentage of INRs within, below and above the range among operated valvular heart disease, with and without AF.

**Figure 3 jcm-12-01141-f003:**
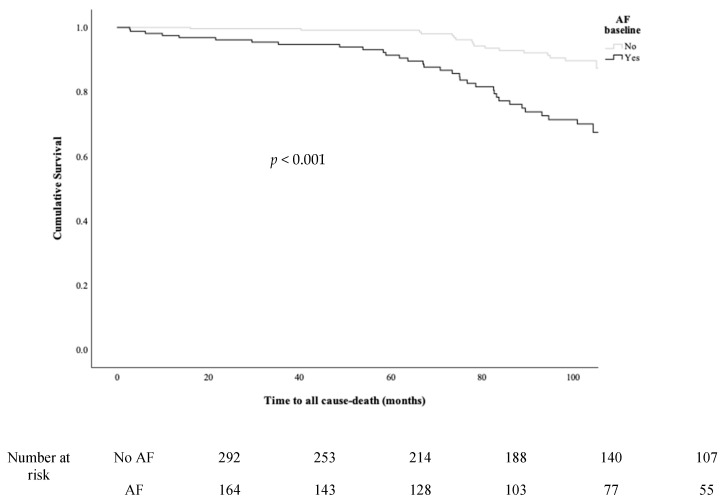
Kaplan–Meier curves among operated VHD patients stratified by the presence of AF for all-cause death.

**Figure 4 jcm-12-01141-f004:**
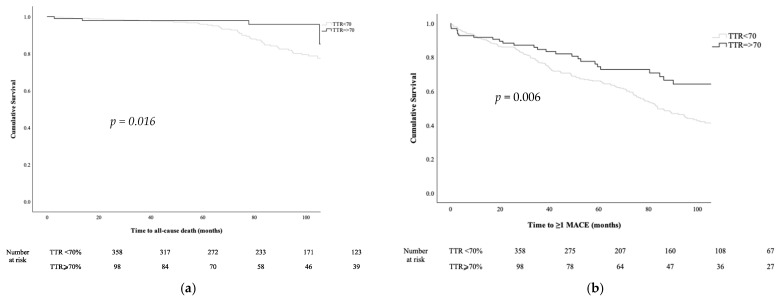
(**a**) Kaplan–Meier curves among operated VHD patients stratified by categories of TTR (TTR < 70% vs. TTR ≥ 70%) for all-cause death. (**b**) Kaplan–Meier curves among operated VHD patients stratified by categories of TTR (TTR < 70% vs. TTR ≥ 70%) for composites of thromboembolic, bleeding event, cardiovascular hospitalisation and all-cause death (≥1 MACE).

**Table 1 jcm-12-01141-t001:** Baseline characteristics of patients with operated valvular heart disease, with and without AF.

*N* (%)		Total *N =* 456	AF *N =* 164	No AF *N =* 292	*p*-Value
Age at implantation	Mean age (SD)	51.1 (14.7)	56.6 (13.3)	48.0 (15.0)	<0.001
Age groups	<65	382 (83.8)	117 (71.3)	265 (90.8)	<0.001
	65–74	59 (12.9)	35 (21.3)	24 (8.2)	
	≥75	15 (3.3)	12 (7.3)	3 (1.0)	
Sex	Female	162 (35.5)	79 (48.2)	83 (28.4)	<0.001
	Male	294 (64.5)	85 (51.8)	209 (71.6)	
Ethnic groups ^‡^ (*N =* 454)	White	296 (65.2)	114 (69.9)	182 (62.5)	0.20
	South-Asian	120 (26.4)	35 (29.2)	85 (29.2)	
	Afro-Caribbean	38 (8.4)	14 (8.6)	24 (8.2)	
Alcohol intake	Alcohol > 14 unit/day (*N =* 353)	32 (9.1)	9 (6.8)	23 (10.4)	0.26
Smoking status	Smoking/ex-smoker (*N =* 372)	83 (22.3)	27 (19.1)	56 (24.2)	0.25
Site(s) of prosthesis	Mitral	110 (24.1)	68 (41.5)	42 (14.4)	<0.001
	Aortic	292 (64.0)	62 (37.8)	230 (78.8)	<0.001
	Both mitral and aortic	54 (11.8)	34 (20.7)	20 (6.8)	<0.001
Types of valve replacement	Mechanical valve	438 (96.1)	150 (91.5)	288 (98.6)	<0.001
	Tissue valve	18 (3.9)	14 (8.5)	4 (1.4)	<0.001
Past medical history	Heart failure	53 (11.6)	35 (21.3)	18 (6.2)	<0.001
	Hypertension	291 (63.8)	118 (72.0)	173 (59.2)	0.007
Past medical history	Diabetes	71 (15.6)	32 (19.5)	39 (13.4)	0.08
	Stroke/TIA	66 (14.5)	30 (18.3)	36 (12.3)	0.08
	Vascular disease *	118 (25.9)	35 (21.3)	83 (28.4)	0.10
	Lung disease ^#^	89 (19.5)	42 (25.6)	47 (16.1)	0.014
	Kidney disease ^†^	17 (3.7)	7 (4.3)	10 (3.4)	0.65
	Anaemia/previous bleeding	189 (41.4)	70 (42.7)	119 (40.8)	0.69
Current medications	Beta-blocker	177 (38.8)	62 (37.8)	115 (39.4)	0.74
	ACEI/ARB	247 (54.2)	94 (57.3)	153 (52.4)	0.31
	Diuretics	233 (51.1)	115 (70.1)	118 (40.4)	<0.001
	Amiodarone	50 (11.0)	37 (22.6)	13 (4.5)	<0.001
	Concurrent mono antiplatelet	79 (17.3)	21 (12.8)	58 (19.9)	0.06
	Digoxin	69 (15.1)	59 (36.0)	10 (3.4)	<0.001
	Calcium channel blocker	54 (11.8)	20 (12.2)	34 (11.6)	0.86
CHA_2_DS_2_-VASc score	Mean (SD)	2.0 (1.4)	2.6 (1.5)	1.7 (1.3)	<0.001
CHA_2_DS_2_-VASc score categories	Low risk	102 (22.4)	21 (12.8)	81 (27.7)	<0.001
	Intermediate	134 (29.4)	45 (27.4)	89 (30.5)	
	High risk	220 (48.2)	98 (59.8)	122 (41.8)	
HAS-BLED score	Mean (SD)	1.6 (1.2)	1.8 (1.1)	1.5 (1.2)	0.014
HAS-BLED score categories	Low risk (0–2)	359 (78.7)	127 (77.4)	232 (79.5)	0.61
	High risk (≥3)	97 (21.3)	37 (22.6)	60 (20.5)	
SAMe-TT_2_R_2_ score	Mean (SD)	2.7 (1.4)	2.7 (1.4)	2.7 (1.4)	0.53
SAMe-TT_2_R_2_ score categories	0–2	200 (43.9)	71 (43.3)	129 (44.2)	0.86
	>2	256 (56.1)	93 (56.7)	163 (55.8)	

ACEI/ARB: angiotensin converting enzyme inhibitor/angiotensin receptor blockade; AF: atrial fibrillation; CHA_2_DS_2_-VASc score—Congestive heart failure/left ventricular dysfunction, Hypertension, Age ≥ 75 years [2 points], Diabetes, Stroke [2 points], Vascular disease, Age 65–74 years, and Sex (female). Total scores range between 0–9; low risk CHA_2_DS_2_-VASc score: 0 male; 1 female, intermediate: 1male, ≥2 female, high risk CHA_2_DS_2_-VASc score: ≥2 male; ≥3 female; TIA: transient ischemic attack; eGFR: estimated glomerular filtration rate, mL/min/1.73 m^2^; HAS-BLED score—uncontrolled Hypertension: systolic ≥ 160 mmHg, Abnormal renal/liver function, Stroke, Bleeding history or predisposition, Labile INR ratio/TTR < 60, Drugs/alcohol concomitantly. Total scores range between 0–9; low risk of bleeding range between 0–2 and high risk of bleeding ≥ 3; SAMe-TT2R2 score—Sex (female), Age < 60, Medical history (more than two comorbidities), Treatment (interacting drug, e.g., Amiodarone), Tobacco use (doubled) and Race (non-white, doubled). Total scores ranged from 0–8; probable good response to VKA therapy range between 0–2 and probable poor response to VKA therapy ranged from ≥3; SD: standard deviation; * vascular disease: prior myocardial infarction, peripheral artery disease or aortic plaque; ^#^ lung disease: obstructive and restrictive diagnosed lung conditions; ^†^ eGFR < 60 mL/min or as noted in medical notes; ^‡^ 2 missing information on ethnicity.

**Table 2 jcm-12-01141-t002:** Measures of anticoagulation control of patients with operated valvular heart disease, with and without AF.

Measures of Anticoagulation Control, *N* (%)	Total, *N =* 456	AF *N =* 164	No AF *N =* 292	*F*-Value	X^2^ Value	*p*-Value
Median (IQR) target INR 2.5 ^‡^	110 (24.1)	33 (20.1)	77 (26.4)	-	6.76	0.034
3.0	157 (34.4)	50 (30.5)	107 (36.6)	-		
3.5	189 (41.4)	81 (49.4)	108 (37.0)	-		
Mean (SD) TTR Rosendaal *	58.5 (14.6)	55.7 (14.2)	60.1 (14.6)	0.09	-	0.002
TTR < 70%	358 (78.5)	141 (86.0)	217 (74.3)	-	8.46	0.004
TTR ≥ 70%	98 (21.5)	23 (14.0)	75 (25.7)	-		
Mean (SD) PINRR *	50.1 (13.8)	47.4 (13.5)	51.6 (13.7)	0.60	-	0.002
PINRR < 70%	417 (91.4)	154 (93.9)	263 (90.1)	-	1.97	0.16
PINRR ≥ 70%	39 (8.6)	10 (6.1)	29 (9.9)	-		
Mean (SD) number of INR tests	96.2 (55.3)	100.7 (58.8)	93.7 (53.1)	0.60	-	0.19
Mean (SD) percentage INRs below the range	25.2 (12.1)	28.4 (12.5)	23.4 (11.6)	0.85	-	<0.001
Mean (SD) percentage above the range	24.9 (9.5)	24.1 (8.6)	25.3 (9.9)	0.64	-	0.22
INR > 5	312 (68.4)	118 (72.0)	194 (66.4)	1.48	-	0.22
INR > 8	64 (14.0)	26 (15.9)	38 (13.1)	-	0.70	0.40
Median (IQR) years of follow-up	6.24 (3.3–8.5)	5.7 (3.7–8.5)	5.7 (3.1–8.5)	-	-	0.87

AF: atrial fibrillation; INR: international normalized ratio; IQR: interquartile range; PINRR: percentage of INRs within range; SD: standard deviation; TTR: time in therapeutic range; * TTR and PINRR were calculated based on the INR ranges obtained from the anticoagulation clinic; ^‡^ Median target INR ranges for each patient were different depending on indication and type of surgery and valve used which was set by the operating surgeon.

**Table 3 jcm-12-01141-t003:** Models of predictors of poor TTR (<70%) in the overall cohort of patients with operated valvular heart disease.

Predictors	Model 1 ^α^ (OR 95% CI)	Model 2 ^†^	Model 3 ^‡^	Model 4 ^¥^	Model 5 ^§^	Model 6 ^¶^
Age (continuous)	1.00 (0.98–1.02); *p* = 0.98	1.00 (0.98–1.02); *p* = 0.96	1.12 (0.94–1.34); *p* = 0.21 ^‡^	-	1.12 (0.93–1.34); *p* = 0.23 ^§^	-
Female sex	1.93 (1.13–3.30); *p* = 0.016	2.05 (1.21–3.50); *p* = 0.008		2.28 (1.29–4.02); *p* = 0.004		2.51 (1.42–4.44); *p* = 0.002
Site of replacement (2 valves vs. 1 valve) *	2.06 (0.77–5.48); *p* = 0.15	1.15 (0.30–4.35); *p* = 0.84 ^†^	2.45 (0.93–6.44); *p* = 0.07	2.02 (0.73–5.58); *p* = 0.17	1.16 (0.31–4.36); *p* = 0.83 ^§^	1.99 (0.50–7.90); *p* = 0.33 ^¶^
Atrial fibrillation	1.75 (1.01–3.03); *p* = 0.045	1.89 (1.10–3.27); *p* = 0.022	1.74 (1.01–3.00); *p* = 0.047	1.38 (0.78–2.43); *p* = 0.26	1.94 (1.13–3.33); *p* = 0.016	1.51 (0.86–2.65); *p* = 0.16
Anaemia/bleeding history	1.84 (1.13–3.00); *p* = 0.014	1.86 (1.14–3.03); *p* = 0.012	1.72 (1.06–2.80); *p* = 0.028	2.60 (1.98–3.43); *p* = <0.001 ^¥^	1.75 (1.08–2.84); *p* = 0.024	2.65 (2.01–3.49); *p* = <0.001 ^¶^

* 2 valves: aortic AND mitral valve vs. 1 valve: aortic OR mitral valve; ^α^ Model 1 includes age, female, site or replacement (2 vs. 1 valve), AF, anaemia/bleeding history; ^†^ Model 2 includes age; female, type of valve (mechanical vs. tissue), AF, anaemia/bleeding history; ^‡^ Model 3 includes CHA_2_DS_2_-VASc score, site or replacement (2 vs. 1 valve), AF, anaemia/bleeding history; ^¥^ Model 4 includes HAS-BLED score, female, site or replacement (2 vs. 1 valve), AF; ^§^ Model 5 includes CHA_2_DS_2_-VASc score, type of valve (mechanical vs. tissue), AF, anaemia/bleeding history; ^¶^ Model 6 includes female, type of valve (mechanical vs. tissue), AF and HAS-BLED score.

**Table 4 jcm-12-01141-t004:** Adverse major clinical events among patients with operated valvular heart disease, with and without AF.

Outcomes, *N* (%)	Total, *N =* 456	Event Rate/100 pt yrs	AF *N =* 164	Event Rate/100 pt yrs	No AF *N =* 292	Event Rate/100 pt yrs	*p*-Value *
Stroke/TIA/SE	25 (5.5)	1.0	8 (4.9)	0.9	17 (5.8)	1.1	0.67
Bleeding *	85 (18.6)	3.6	30 (18.3)	3.6	55 (18.8)	3.6	0.89
CV hospitalisation	78 (17.1)	3.4	31 (18.9)	3.8	47 (16.1)	3.2	0.45
All-cause death	51 (11.2)	1.9	34 (20.7)	3.6	17 (5.8)	1.0	<0.001
CV death	12 (2.6)	0.5	12 (7.3)	1.3	0	-	<0.001
Non-CV death	39 (8.6)	1.5	22 (13.4)	2.3	17 (5.8)	1.0	0.009
≥1 MACEs ^†^	180 (39.5)	8.7	75 (45.7)	10.1	105 (36.0)	7.8	0.051

Cardiovascular hospitalisation: a hospitalisation with a cardiovascular cause: (i) heart failure, MI, new angina, non-fatal cardiac arrest, ventricular arrhythmia, uncontrolled AF/atrial flutter, supraventricular arrhythmia, (ii) valve surgery, CABG surgery, PTCA surgery, pacemaker/ICD insertion, carotid endarterectomy, peripheral angioplasty/surgery, limb amputation AND as recorded in patient’s medical documents; DVT—Deep Vein Thrombosis; * Bleeding ISTH is a combination of major bleed ISTH and clinically relevant non-major bleed (CRNMB); Major Bleeding—ISTH Major Bleeding: fatal bleeding and/or symptomatic bleeding in a critical area or organ, such as intracranial, intraspinal, intraocular, retroperitoneal, intraarticular or pericardial, or intramuscular with compartment syndrome and/or bleeding causing a fall in haemoglobin level of 2 g/dL (1.24 mmol/L) or more, or leading to transfusion of two or more units of whole blood or red cells; Clinically relevant non-major bleeding (CRNMB): clinically overt bleeding not satisfying the criteria for major bleeding but meeting at least one of the 3 criteria: (i) leading to hospitalisation or increased level of care, (ii) requiring medical intervention by healthcare professional and (iii) prompting face to face evaluation; SE: systemic embolism; TIA: transient ischemic attack; VTE: venous thromboembolism. ^†^ ≥1 MACEs: adverse clinical event defined a composite of TE, bleeding, CV hospitalisation and all-cause death; * *p*-value for proportion.

## Data Availability

Data sharing is restricted due to institutional policies.

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
