# Peer review of "Anticoagulation Control and Major Adverse Clinical Events in Patients with Operated Valvular Heart Disease with and without Atrial Fibrillation Receiving Vitamin K Antagonists"

_jcm, 2023, doi:10.3390/jcm12031141_

Round 1
Reviewer 1 Report
In their manuscript „Anticoagulation control and major adverse clinical events in patients with operated valvular heart disease with and without atrial fibrillation receiving vitamin K antagonists“, Zulkifly et al. investigated the quality of anticoagulation control and occurrence of major adverse clinical events in patients with operated valvular heart disease (VHD) with and without atrial fibrillation (AF) receiving vitamin K antagonists. They found that operated VHD patients with AF have poorer anticoagulation control and a lower survival rate compared to those without AF and that the presence of concomitant AF, anaemia/bleeding history and female sex independently predict poor TTR.
Overall, this is a nice study with a long follow-up discussing a relevant topic that has not been adequately assessed in literature yet.
1. Please delete company names from the introduction in line 52 and shorten the last paragraph of the introduction, as it is not necessary to mentions all aims (especially including all outcomes that were investigated) in such detail.
2. One limitation is that about 1/6 of the initial study population were excluded due to missing medical information. This might represent a selection bias.
3. Please shorten paragraphs 2.4 and 2.5, as it is not necessary to explain these information in such detail.
4. There might be a difference regarding baseline characteristics and clinical outcomes between patients with presence of known AF at the time of surgery vs. those with AF diagnosed post-operatively (as assessed in line 154). Please discuss. How many patients were in each of these groups?
5. How many patients had paroxysmal, persistent or permanent AF? Were there any differences between groups?
6. If patients presented with AF at the time of surgery, when was AF diagnosed for the first time? Were these patients treated with electrocardioversion or catheter ablation in the past? E.g., a long history of AF might have affected clinical outcomes.
7. Patients with AF were older and had more comorbidities than those without AF. This might be a confounding factor for clinical outcomes. Please discuss.
8. Why did you mention patients with INR >5 and >8 and did not choose different cut-offs?
9. The here presented data are from 2017-2018. Anticoagulation control strategies might have changed and improved. Please discuss.
10. Figure 2a: It is not optimal to have mean TTR and PINRR on the one hand and the proportion of patients with TTR≥70% on the other hand in one Figure with the same y axis. Please revise.
11. Please include a p-value in Figure 4b and numbers at risk in all Kaplan-Meier analyses. Please improve inconsistencies in the style of the presented figures (same font size, labeling of x/y axis etc.).
12. The quality of anticoagulation control was lower in operated VHD patients with AF compared to those without AF. What exactly may be a reason for that? Please discuss in a greater detail.
13. Please explain “several factors” in line 109. Why may female sex predict poor TTR?
14. Please review the language style of the manuscript thoroughly: In detail, please only use whole sentences (e.g., in the Results section of the Abstract and at the beginning of the Methods section in the text). Please exclude the terms “SD” and “IQR” in the Abstract and the manuscript text. Please introduce abbreviations at first use (e.g., VHD in line 43, LVEF in line 44, TTR in line 60).
Author Response
Thank you for reviewing our paper and the opportunity to revise the manuscript. We would like to submit our revised manuscript and we have prepared an itemised response to the reviewers on the next page.
1. Please delete company names from the introduction in line 52 and shorten the last paragraph of the introduction, as it is not necessary to mentions all aims (especially including all outcomes that were investigated) in such detail.
The company names have been deleted from the Introduction.
The last paragraph has been shortened and now reads:
“Various studies [6-12] conducted between 2002 and 2021, have investigated anticoagulation control after valve replacement; two studies [8-10] used anticoagulation variability while the others [6, 7, 9, 10] used time in therapeutic range (TTR). Therefore, the objectives of this study were to investigate anticoagulation control measured using TTR (Rosendaal method) and the PINRR method among operated patients with VHD among those with and without AF and to investigate the predictors of poor anticoagulation control (TTR<70%) and the prevalence of major adverse clinical events (MACE).”
2. One limitation is that about 1/6 of the initial study population were excluded due to missing medical information. This might represent a selection bias.
This limitation has now been acknowledged in the Discussion and reads:
This is the first study investigating anticoagulation control in the UK among operated VHD patients stratified by the presence of AF at baseline (obtained from the post-operative notes). Although it is limited by the relatively small sample size (with approximately one-sixth of the eligible cohort excluded due to missing medical information), it provides some insights on anticoagulation control among operated VHD patients, with and without AF.
3. Please shorten paragraphs 2.4 and 2.5, as it is not necessary to explain these information in such detail.
Paragraphs 2.4 and 2.5 have been significantly shortened. The following information has been deleted.
“Information on smoking status was available for 372 patients (82%); data on alcohol intake were only available for 353 patients (77%). This information was used to calculate the individual HAS-BLED and SAMe-TT2R2 scores”.
“Assumptions were made for presence/absence of chronic kidney disease (serum creatinine >200umol/L), liver disease (ALT/ALP >x3 ULN) and anaemia (as haemoglobin level of <135 g/L for male and <115 g/L for female) based on the laboratory results.”
“Stroke was defined as any focal neurologic deficit, from a non-traumatic cause, lasting at least 24 hours and further categorized as ischaemic (with or without haemorrhagic transformation), haemorrhagic, or of uncertain type (where brain imaging or autopsy was not performed). A thromboembolic event outside the brain, retina, heart or lungs was classified as systemic embolism.”
Definition of CV hospitalisation: “i) heart failure, myocardial infarction (MI), new angina, non-fatal cardiac arrest, ventricular arrhythmia, uncontrolled AF/atrial flutter, supraventricular arrhythmia, ii) valve surgery, coronary artery bypass graft (CABG) surgery, percutaneous transluminal coronary angioplasty (PTCA) surgery, pacemaker/ implantable cardioverter-defibrillator (ICD) insertion, carotid endarterectomy, peripheral angioplasty/surgery, limb amputation.”
Definitions of major and CRNMB was also deleted “major bleeding: fatal bleeding, and/or symptomatic bleeding in a critical area or organ, such as intracranial, intraspinal, intraocular, retroperitoneal, intra-articular or pericardial, or intramuscular with compartment syndrome, and/or bleeding causing a fall in haemoglobin level of 20 g/L or more, or leading to transfusion of two or more units of whole blood or red cells) (1). Clinically relevant non-major bleeding (CRNMB) was defined as clinically overt bleeding that did not satisfy the criteria for major bleeding and that led to hospital admission, physician-guided medical or surgical treatment, or a change in antithrombotic therapy (1).
4. There might be a difference regarding baseline characteristics and clinical outcomes between patients with presence of known AF at the time of surgery vs. those with AF diagnosed post-operatively (as assessed in line 154). Please discuss. How many patients were in each of these groups?
Thank you for your question. There was a total of 164 AF patients overall. Of these, only 18 patients (3.9%) were given the diagnosis of AF post-operatively compared to 146 patients (32%) with known AF at baseline. There was no difference in age [56.5 (13.6) vs. 57.9; p=0.671] and other relevant demographic and clinical parameters. Operated VHD patients with AF at baseline were more likely to be female (51.4 vs 22.2; p=0.024), have had their mitral valve (45.9% vs. 5.6%; p=0.001) or lower aortic valve (32.9 vs.77.8%; p<0.001) replaced, more likely to have concomitant hypertension (74.7% vs, 50%; p=0.028) and prescribed with amiodarone (61.1% vs. 17,8%) and digoxin (40% vs. 0; p<0.001) compared to those diagnosed with AF postoperatively.
We have not included this detailed information in the manuscript as the comparison between known AF and post-operative AF was 146 vs. 18 patients and the small number of people in the latter group may have influenced the significant differences in demographic and clinical characteristics.
In terms of anticoagulation control, operated VHD patients with AF at baseline were more likely to have poorer mean TTR (Roseendal and PINRR Method) [54.9 (14.2) vs 62.0 (12.8); p=0.047] and [46.5 (14.0) vs. 54.5 (14); p=0.028] respectively and higher proportion of INRs below target range (29.2 vs. 21.4; p=0.02). There is no difference in any of the adverse clinical outcome of interest.
5. How many patients had paroxysmal, persistent or permanent AF? Were there any differences between groups?
Thank you for your question. There were 38/164 (23.2%), 32/164 (19.5%) and 76/164 (46.3%) of patients with paroxysmal, persistent and permanent AF respectively. This has been added (lines 330-331).
6. If patients presented with AF at the time of surgery, when was AF diagnosed for the first time? Were these patients treated with electrocardioversion or catheter ablation in the past? E.g., a long history of AF might have affected clinical outcomes.
Thank you for your question. Unfortunately, data on whether patients presented with AF at the time of surgery and it’s treatment was not obtained during the time of data collection (2018).
7. Patients with AF were older and had more comorbidities than those without AF. This might be a confounding factor for clinical outcomes. Please discuss.
Thank you for your suggestion. Age and comorbidities were incorporated in the CHA2DS2-VASc score and included in Model 3 (Table 3) as independent predictor of poor TTR in the overall cohort. However, in Model 3, the CHA2DS2-VASc score did not independently predict poor TTR [OR 1.12 (0.94-1.34); p=0.21] instead the presence of AF [1.74 (1.01-3.00); p=0.047] and underlying anemia/bleeding history [1.72 (1.06-2.80); p=0.028] did predict poor TTR (TTR<70).
8. Why did you mention patients with INR >5 and >8 and did not choose different cut-offs?
Thank you for your question INR >5 and >8 were chosen as the cut off as it represent supratherapeutic INRs and it is an indicator of poor anticoagulation control apart from TTR <65% by the NICE guideline on AF: Diagnosis and Management (2). This sentence has been added at line 209-211 and reads:
“The proportions of sub-therapeutic INRs (INRs below the target range), supra-therapeutic (INRs above target range) and patients with at least one INR >5.0 or >8.0 were also collected as these are also indicator of poor anticoagulation control. The risk of bleeding also increases when the INR exceeds 4, and the risk rises sharply with values >5 and >8 respectively(3).
However, in the current population, there is no significant difference in the proportion of those with and without AF with at least one INR reading of >5 and >8 respectively.”
9. The here presented data are from 2017-2018. Anticoagulation control strategies might have changed and improved. Please discuss.
Thank you for your suggestion. However, anticoagulation strategies and recommendations for patients with mechanical heart valves have not changed since the data presented here were analysed. VKAs are still the recommended treatment strategy with the same target INRs as in 2018.
10. Figure 2a: It is not optimal to have mean TTR and PINRR on the one hand and the proportion of patients with TTR≥70% on the other hand in one Figure with the same y axis. Please revise.
Thank you for your comment. The proportion of TTR>70% has been removed from Figure 2a
11. Please include a p-value in Figure 4b and numbers at risk in all Kaplan-Meier analyses. Please improve inconsistencies in the style of the presented figures (same font size, labeling of x/y axis etc.).
Thank you for your comments. The figures have been revised accordingly.
12. The quality of anticoagulation control was lower in operated VHD patients with AF compared to those without AF. What exactly may be a reason for that? Please discuss in a greater detail.
Operated VHD patients with AF are older and have multiple comorbidities with complex disease management which might contribute to the lower quality of anticoagulation control (4-9).” Furthermore, in this cohort of operated VHD patients with AF were more likely to have VHD at mitral site, which requires a higher INR target than NVAF patients or patients with VHD at aortic site which may be more difficult to achieve and maintain.the This has been added in lines 105-107
13. Please explain “several factors” in line 109. Why may female sex predict poor TTR?
Thank you for your suggestion. The text at lines 113-116 has been amended to read:.
“The finding that being female predicts poor TTR is consistent with other non-valvular AF studies (4, 6, 7, 10-13). One large American study (14) evaluating medication use and adherence among 16.0 million women and 13.5 million men showed that women were more likely to be non-adherent to their diabetic (35.4% vs. 32.5%; p<0.0001) and antihypertensive (25.8% vs. 24.8; p<0.0001) medications compared to men respectively and also speculated due to more complex medications regime, more side effects and more responsibilities resulting in self neglect compared to men (14).”
14. Please review the language style of the manuscript thoroughly: In detail, please only use whole sentences (e.g., in the Results section of the Abstract and at the beginning of the Methods section in the text). Please exclude the terms “SD” and “IQR” in the Abstract and the manuscript text. Please introduce abbreviations at first use (e.g., VHD in line 43, LVEF in line 44, TTR in line 60).
Thank you. Whole sentence was used in abstract except for AF, TTR and PINRR as it is the standard abbreviation used throughout the text. Terms SD and IQR were removed in abstract and text. All abbreviations were introduced at first use.
Reviewer 2 Report
Interesting and important topic, but I would not recommend a modification of the main focus, the investigation is not about the quality of anticoagulation. Since the data is based on only three INR values after surgery.

Author Response
Thank you for reviewing our paper and the opportunity to revise the manuscript. We would like to submit our revised manuscript and we have prepared an itemised response to the reviewers on the next page.
Interesting and important topic, but I would not recommend a modification of the main focus, the investigation is not about the quality of anticoagulation. Since the data is based on only three INR values after surgery.
Thank you for your feedback. In the methods, it is stated that data is based on at least 3 INR readings will be used to calculate the TTR. However, the mean (SD) number of INR reading that were used to calculate TTR in this study is 96.2 (55.3) over a median period of 624 (3.3-8.5) years of follow up. This result can be seen in Table 2.
Abstract
1. Line 16- to add ischemic and thromboembolic complications.
This has been added.
2. Line 20- quality of anticoagulation might not be the adequate term, considering that only very few INR values were registered per patient. Rather a point-of-care measuring (CoaguChek), daily INR values would be a good marker for high quality anticoagulation.
The TTR was based on at least 3 INR readings. However, the mean (SD) number of INR reading that were used to calculate TTR in this study is 96.2 (55.3) over a median period of 624 (3.3-8.5) years of follow up. this result can be seen in Table 2. Therefore, we consider quality of anticoagulation as our main objective.
Introduction
1. Line 58-which patients with tissue valves are requiring anticoagulation in your department? e.g. in our department only mitral valve tissue valves need anticoagulation, for aortic position only mono-APT is required for three months (in absence of AF)
Thank you for your question. There were only 18 patients (3.9%) with tissue valves required anticoagulation with VKA in this cohort. Of those, 14 (8.5%) had AF at baseline. This information is available in Table 1.
Method
1. Line 133- only INR or in relation to liver parameters or signs of infection?
The year 2009 was used as a baseline when INR values consistently available throughout the hospital system. Other baseline and lab parameters were collected as near to the date (or within one month) of VKA initiation after the first valve surgery.
2. line 144-where there more INR measurements perfomed at all? At the GP? Self-measurements?
There was a mean of 96.2 (55.3) INR readings collected from the electronic health records available at the Sandwell West Birmingham Hospitals. All INR readings for a given patient were captured as GP systems were connected to the hospital INR system. No patients were self-measuring INR.
Results
1. Line 215-which type of valves have been used? (company, prosthesis type)
Data on company and specific type of valves were not collected.
2. Line 220-but only AF or non-AF at the time of surgery was evaluated? Or were there newer informations/ECG?
Both VHD patients with and without AF was evaluated at the time of surgery. No new information/ECG was obtained at the time of data collection (2018).
3. Table 1: mono or dual or any antiplatelet medication?
Only mono-antiplatelet medication was recorded. This has been added to Table 1.
Discussion
1. Line 92: again, quality of anticoagulation is in my opinion not the right term and not what you have evaluated. Rather its anticoagulation status in patients with history of heart valve replacement and atrial fibrillation or normal sinus rhythm at the time of surgery.....
We believe that quality of the anticoagulation is correct. In response to earlier comments, the data is based on at least 3 INR readings used to calculate the TTR. However, the mean (SD) number of INR reading that were used to calculate TTR in this study is 96.2 (55.3) over a median period of 6.24 (3.3-8.5) years of follow up; seen in Table 2. Hence, we will consider quality of anticoagulation as our main objective.
Round 2
Reviewer 1 Report
Altogether, the comments have been properly addressed. The manuscript has substantially improved.